# Point prevalence of asymptomatic Plasmodium infection and the comparison of microscopy, rapid diagnostic test and nested PCR for the diagnosis of asymptomatic malaria among children under 5 years in Ghana

**Bismark Okyere[1]☯, Alex Owusu-Ofori[1]☯, Daniel Ansong[2‡], Rebecca Buxton[3‡], Scott Benson[4‡], Alex Osei-Akoto[2‡], Eddie-Williams Owiredu[5‡], Collins Adjei[1‡], Evans Xorse Amuzu[6‡], Joseph Marfo Boaheng[6‡], Ty Dickerson[7]☯\***

1 Department of Clinical Microbiology, School of Medicine and Dentistry, Kwame Nkrumah University of Science and Technology, Kumasi, Ghana, 2 Department of Child Health, School of Medicine and Dentistry, Kwame Nkrumah University of Science and Technology, Kumasi, Ghana, 3 Medical Laboratory Science Division, Department of Pathology, University of Utah School of Medicine, Salt Lake City, Utah, United States of America, 4 Division of Public Health, Department of Family and Preventive Medicine, University of Utah School of Medicine, Salt Lake City, Utah, United States of America, 5 Department of Molecular Medicine, School of Medicine and Dentistry, Kwame Nkrumah University of Science and Technology, Kumasi, Ghana, 6 Research and Development Unit, Komfo Anokye Teaching Hospital, Kumasi, Ghana, 7 Division of Pediatric Inpatient Medicine, Department of Pediatrics, University of Utah School of Medicine, Salt Lake City, Utah, United States of America

☯ These authors contributed equally to this work.
‡ These authors also contributed equally to this work.
\* ty.dickerson@hsc.utah.edu

## Abstract

### Background

*Plasmodium* infection among children is a serious public health problem. Asymptomatic malaria infection among humans serves as a significant reservoir for transmitting *Plasmodium* to uninfected *Anopheles* mosquitoes, fueling malaria endemicity and asymptomatic malaria may progress to clinical malaria. Therefore, prompt and accurate diagnosis of malaria infection is crucial for the management and control of malaria, especially in endemic areas. This study assessed the point prevalence of asymptomatic malaria infection and evaluated the performance of malaria Rapid Diagnostic Tests (RDT), light microscopy and nested PCR (nPCR) for the diagnosis of asymptomatic malaria infection in a paediatric population in the Atwima Nwabiagya North district, Ghana.

### Methods

This cross-sectional study enrolled 500 asymptomatic children aged $\leq$ 5 years. After consent was obtained from a parent, blood samples were collected from each participant to

**Data Availability Statement:** All relevant data are within the manuscript and its Supporting Information files.

**Funding:** All authors received support via salary from their respective institutions: University of Utah School of Medicine (TD, RB, SB), Kwame Nkrumah University of Science and Technology School of Medical Sciences (BO, AOO, DA, AOA, CA, EWO), and Komfo Anokye Teaching Hospital (EXA, JMB). The authors received no other specific funding for this study.

**Competing interests:** The authors have declared that no competing interests exist.

assess for *Plasmodium* infection based on histidine rich protein-2 (*pf*HRP-2)-based malaria RDT, light microscopy and nPCR.

## Results

The point prevalence of asymptomatic malaria by microscopy, RDT, and nPCR were 116/ 500 (23.2%), 156/500 (31.2%), and 184/500 (36.8%), respectively. Using nPCR as the reference, RDT presented with a perfect sensitivity (100.0%), specificity (100.0%), accuracy (100.0%), and reliability (100.0%) in detecting asymptomatic *P. falciparum* infection. Likewise, microscopy presented with an excellent specificity and high accuracy in detecting both *P. falciparum* (100.0%; 85.6%) and *P. malariae* (100.0%; 100.0%). However, the sensitivity (56.4%) and reliability (56.4%) of microscopy was low for both *P. falciparum*.

## Conclusion

The findings of this study indicate a high point prevalence of asymptomatic *Plasmodium* infection among children in Atwima Nwabiagya North district, Ghana. In the absence of the more sensitive PCR, pfHRP-2-based malaria RDT provides substantial diagnostic sensitivity, specificity, accuracy and reliability and is superior to microscopy.

## Introduction

Malaria is a pervasive parasitic disease in the tropical and subtropical regions which is mostly prevalent in sub-Saharan Africa, Asia, and Latin America [1]. Currently, the World Health Organization (WHO) estimates 219 million cases and 435,000 malaria-related deaths globally [2]. In the WHO African Region, malaria causes significant morbidity and mortality with annual infection and mortality rates of 213 million and 380,000 individuals, respectively, and it claims the life of a child under five years every two minutes [3, 4]. Despite successes in global malaria control in previous years, recent data indicate insufficient progress. In Ghana, malaria remains a major cause of loss of days of healthy life, accounting for not less than 20% of child deaths, 40% of child hospital admissions, and more than 50% of outpatient attendances [5–8]. The enormous toll on life and both national and household economics [9] underscores the need for ongoing malaria diagnosis, treatment, and disease surveillance.

Clinically, the diagnosis of malaria is often based on signs and symptoms alone. However, due to overlapping symptoms between malaria and other infectious conditions, a malaria diagnosis based solely on signs and symptoms may be inaccurate leading to improper use of antimalarial medication or the delay in proper diagnosis and treatment of an alternative condition [10]. As a result, the WHO recommends the use of microscopy or rapid diagnostic tests (RDT) as confirmatory diagnostic tools for malaria prior to initiation of treatment in suspected malaria cases, which also minimizes the likelihood of the development of drug resistant strains [11].

In many developing countries, microscopic examination of Giemsa-stained blood smears is considered the "gold standard" for malaria diagnosis and a mandatory test prior to antimalarial therapy. Though it is cost-effective, malaria microscopy is limited by several factors including quality control, limited availability of microscopes, time consuming for optimal film preparation, examination, and interpretation, diagnostic biases as a result of its dependence on operator's experience and low diagnostic sensitivity [12–14]. Furthermore, bacteria, fungi,

dirt, cell debris, and poor blood film preparation result in formation of artifacts and are associated with false positive results [15].

In an effort to improve diagnostic sensitivity and turnaround time and abate diagnostic errors related to microscopy, RDTs were developed. Currently, the most widely utilized RDTs exploit the presence of *Plasmodium falciparum* Histidine-Rich Protein-2 (*pf*HRP-2), parasite-specific Lactate Dehydrogenase (pLDH) or *Plasmodium* aldolase to detect parasitemia [16, 17]. The performance of RDT is influenced by manufacturing and environmental conditions in addition to its inability to quantify parasitemia and to accurately identify species other than *P. falciparum* [18–20]. Additionally, false negatives due to *pf*HRP-2 gene deletions and non-reliability for non-*Plasmodium falciparum* infections have been reported [21–23]. Moreover, persistence of *pf*HRP-2 antigens in circulation even after parasite clearance may result in false positive results; limiting RDT specificity [24].

Taken together, both RDT and microscopy are limited by low detection threshold, especially in low parasitaemic cases [20, 25]. As such, Polymerase Chain Reaction (PCR) based assays have been developed to remedy some of the limitations. PCR is very sensitive, particularly in cases of low density or mixed infection and it is valuable for accurate collection of malaria epidemiological data [26, 27]. However, the expensive, technical and time-consuming nature of PCR limits its utilization in routine practice, especially in remote or resource-limited settings.

Asymptomatic carriers of malaria parasites have recently received considerable attention as global strides towards malaria eradication are underway. Some reports indicate that asymptomatic malaria infection may serve as a significant reservoir, transmitting *Plasmodium* to uninfected *Anopheles* mosquitoes which fuels malaria endemicity [28, 29]. There is also the possibility that asymptomatic malaria will transition into clinical malaria. Thus, accurate diagnosis of asymptomatic malaria as a potential reservoir of infection, especially in children, is crucial. Although a number of studies on asymptomatic malaria in older children have been conducted across Ghana [30–32] and children under 5 in neighboring African countries [33–35], there remains a dearth of published data on asymptomatic malaria in children under 5 years in Ghana, particular in the northern sectors of the country where adequate health facilities are wanting. This study assessed the point prevalence of asymptomatic malaria infection and evaluated the performance of malaria RDT, light microscopy and nested PCR (nPCR) for the diagnosis of asymptomatic malaria infection in children under 5 years old in Atwima Nwabiagya North district, Ghana.

## Materials and methods

### Study design/area and participants

The study was conducted in July, 2015 in peri-urban and rural communities of the Atwima Nwabiagya North district in the Ashanti region of Ghana. The district lies approximately on latitude 6˚ 32'N and 6˚ 75'N and between longitude 1˚ 45' W and 2˚ 00' W. It is located in the western part of the region and shares common boundaries with Offinso Municipal (to the North), Ahafo Ano South and Atwima Mponua Districts (to the West), Amansie-West and Atwima Kwanwoma Districts (to the South), Kumasi Metropolis and Afigya Kwabre Districts (to the East). It covers an estimated area of 294.84 square kilometers and has an estimated population of 149,025 according to the 2010 Population and Housing Census [36].

The minimum sample size of 196 was calculated at 95% confidence level, 7% margin of error, and a response distribution of 50% using the Raosoft sample size calculator [37]. However, in an effort to enhance the statistical power of the study, a total of 500 asymptomatic children of age ≤ 5 years old were recruited to the study. Inclusion criteria were: lack of fever in

the last 3 days, no history of anti-malarial treatment in the last 14 days, an axillary temperature less than 37.5˚C, and no known acute/chronic or disease. Ethical approval for this study was obtained from the committee on Human Research, Publications and Ethics (CHRPE) of the School of Medical Sciences of the Kwame Nkrumah University of Science and Technology (CHRPE/AP/257/15). Written informed consent was obtained from parents/ guardians of all participating children after the aims and objectives of the study had been explained to them. Participation was voluntary, and respondents were assured that the information obtained was strictly for research and academic purposes only and were guaranteed the liberty to opt out from the study at their own convenience.

## Data and sample collection

Parents/guardians of participating children were interviewed to obtain participants' information on gender, age, history of malaria/fever and treatments, and presence of acute and/or chronic disease. Fingerprick blood samples were obtained from children who satisfied the inclusion criteria. About 8 μl of the blood was used for haemoglobin measurement using HemoPoint H2 Hemoglobin analyzer (Accuracy of 14.0 g/dl ± 0.3 g/dl; Linearity of 0–20 g/dl ± 0.3 g/dl; total precision CV <1.5%) (EKF Diagnostics, Stanbio Laboratory, USA). Anaemia was defined as haemoglobin level <11 g/dl, and graded as mild (10–10.9 g/dl), moderate (7–9.9 g/dl), and severe (<7 g/dl) [38]. Approximately 5 μl of blood was used for malaria diagnosis by RDT. Giemsa-stained thick and thin blood films were also prepared. Additionally, about 3–5 drops of the blood were spotted onto Whatman 903™ Filter Paper (Schleicher and Schuell BioScience, Inc., Keene, New Hampshire), air dried and individually kept in sealed plastic bag for subsequent nPCR analysis. All samples were tested for malaria by RDT, microscopy, and nested PCR. PCR was considered the gold standard.

## Malaria diagnosis by RDT

Malaria RDT diagnosis was based on the detection of Histidine rich protein 2 (HRP-2) produced by *P. falciparum* only (paraHIT *f*, Span Diagnostics Limited, Surat, India). Testing and reporting was done according to the manufacturer's instructions. Briefly, approximately 5 μl of the blood sample was transferred to the sample window using a micropipette, followed by 4 drops of the Reaction buffer into the buffer window. A sample was considered positive for *P. falciparum* malaria if the test line and control line appeared within the result window. The presence of the control line only, was considered a negative result. Results were declared invalid if the control line failed to appear within the result window, warranting re-testing. The test was done in duplicates. A third RDT was performed in the case of non-concordant result. Patients who tested positive were referred to the closest community health care facility for further diagnosis and treatment in accordance with the Ghana Health Service guidelines.

## Malaria diagnosis by microscopy

Thick and thin blood smears were prepared (in duplicate) on clean, grease-free glass microscope slides immediately after sample collection. The films were allowed to air-dry and thin films were fixed with methanol. Both thick and fixed thin films were stained with 5% Giemsa solution for 30 minutes prior to microscopic examination. Examination and reporting of both thick and thin films were performed independently by two trained microscopists. The thin film was used to identify the specific species of *Plasmodium*. A film was considered positive by microscopy when both microscopists recorded a positive result for the same species. A film was considered negative only after observing at least 200 high-power fields (HPF) without finding parasites on a thick film. In the case of non-concordant result, a third examination was

performed by a different microscopist. All microscopists were blinded to the results of RDT. Parasites were counted per 200 white blood cells (WBCs) per HPF from the thick film. The parasite density was calculated by assuming a WBC count of 8000/μl and 4.5 million RBC/μl in accordance with the WHO standard [39].

## DNA extraction and molecular analysis

DNA isolation from Whatman filter papers was based on the Chelex-based technique as previously described [40]. Nested polymerase chain reaction was used for the determination of *Plasmodium* species, as previously described [41]. Briefly, *Plasmodium* genus was detected based on amplification of the outer genus-specific primers (rPLU1 and rPLU5). The reaction mixture for the initial outer reaction contained 4 mM of $MgCl_2$, 200 μM DNTPs, 0.0625 μM of each primer and one unit of Taq DNA polymerase (Sigma-Aldrich, USA). For the primary reaction (Nested 1), the PCR cycling conditions consisted of an initial denaturation at 94˚C for 4 min, denaturation at 94˚C for 30 secs, annealing at 60˚C for 1 min, extension at 72˚C for 1 min (for 5 cycles), followed by denaturation at 94˚C for 30 secs, annealing at 55˚C for 1 min, extension at 72˚C for 1 min, and final extension at 72˚C for 4 min (for 45 cycles). Subsequently, a secondary amplification reaction (Nested 2) using the genus-specific (rPLU3 and rPLU4) and species-specific primer pairs (rFAL1 and rFAL2, rMAL1 and rMAL2, rOVA1 and rOVA2) was performed with 1 μL of the product of the first amplification reaction as a template DNA as previously described [27, 42–44] (**Table 1**). All PCR reactions were performed using a GeneAmp PCR System 2700 (Applied Biosystems Incorporated, USA). The amplified products were separated by electrophoresis on 2% agarose gels, stained with 0.5 μg/mL ethidium bromide and visualized under UV light. (**Fig 1**).

## Statistical analysis

Data processing was done using Microsoft Excel 2016. Statistical analysis and graphical presentation was performed using the R Language for Statistical Computing version 3.5.2 (R Core Team, Vienna, Austria) [45]. Categorical data were presented as frequency (percentages). Continuous data were presented as mean ± standard deviation (SD). Univariate logistic regression analysis was used to assess the association between sociodemographic characteristics and malaria infection for each test methods used in this study. The receiver operating characteristics (ROC) curve analysis was used to assess the diagnostic performance of malaria RDT and microscopy using PCR as the reference. Reliability was expressed as the J index [(TP×TN)—

**Table 1. Nested PCR protocol and *Plasmodium* ssrRNA genes used in this study.**

| Target species | Primer | Sequence (5′-3′) | Reaction |
|---|---|---|---|
| *Plasmodium* genus-specific | rPLU1 | TCAAAGATTAAGCCATGCAAGTGA | Nested 1 |
| | rPLU5 | CCTGTTGTTGCCTTAAACTTC | |
| | rPLU3 | TTTTTATAAGGATAACTACGGAAAAGCTGT | Nested 2 |
| | rPLU4 | TACCCGTCATAGCCATGTTAGGCCAATACC | |
| *Plasmodium* species-specific | | | |
| *Plasmodium falciparum* | rFAL1 | TTAAACTGGTTTGGGAAAACCAAATATATT | Nested 2 |
| | rFAL2 | ACACAATGAACTCAATCATGACTACCCGTC | |
| *Plasmodium malariae* | rMAL1 | ATAACATAGTTGTACGTTAAGAATAACCGC | Nested 2 |
| | rMAL2 | AAAATTCCCATGCATAAAAAATTATACAAA | |
| *Plasmodium ovale* | rOVA1 | ATCTCTTTTGCTATTTTTTAGTATTGGAGA | Nested 2 |
| | rOVA2 | GGAAAAGGACACATTAATTGTATCCTAGTG | |

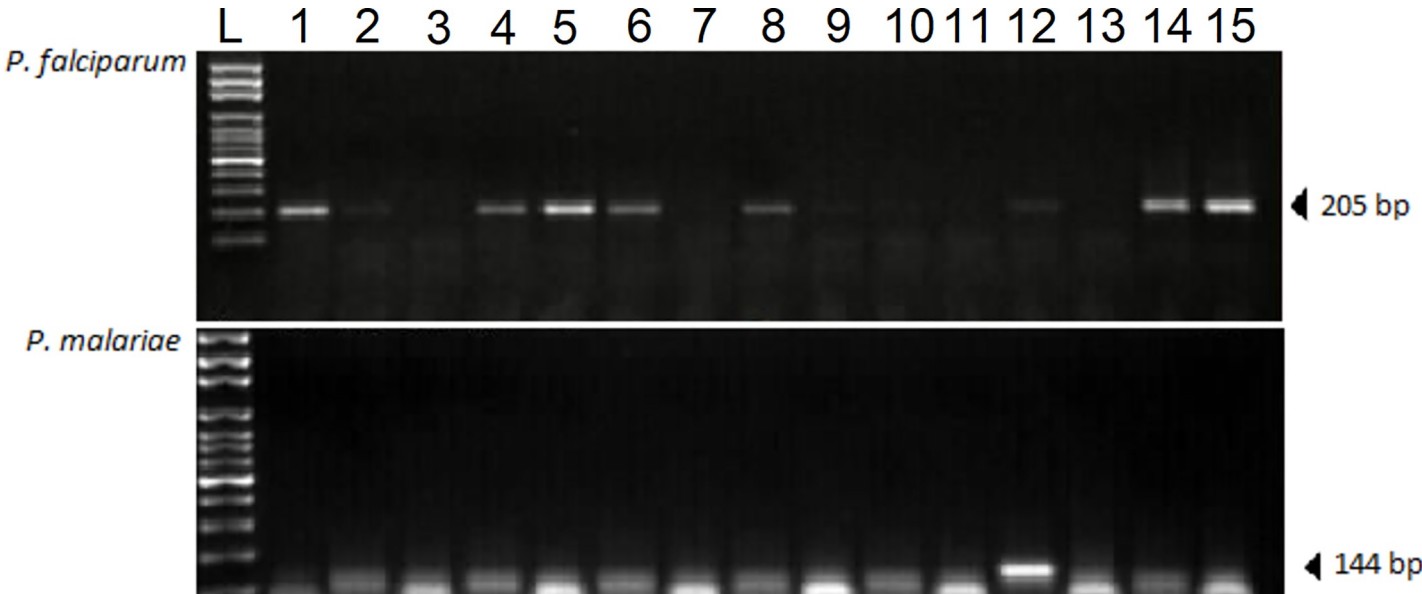

**Fig 1. Agarose gel electrophoresis showing nPCR products under UV light.** The top panel shows representative results of the *P. falciparum*-specific nested PCR. Lanes 1, 4, 5, 6, 8, 12, 14, and 15 contain the positive 205 bp PCR products expected. The bottom panel shows representative results of the *P. malariae*-specific nested PCR. Lane 12 contains the positive 144 bp PCR product expected. Smaller bands are likely due to low DNA concentration in those samples. L represents the molecular ladder.

(FP×FN)]/ [(TP+FN) (TN+FP)]. All tests were two-sided and a *p*-value < 0.05 was considered statistically significant.

## Results

A total of 500 children with mean age and haemoglobin level of 2.21 years and 10.26 g/dl, respectively, were included in this study. A higher proportion of the participants were male (51.2%), between 1–2 years of age (47.2%), and resided in the community of Barekuma (28.4%). A total of 189 (75.6%) were anaemic and the point prevalence of mild, moderate, and severe anaemia was 55.2%, 16.8%, and 3.6%, respectively. Only *P. falciparum* [parasite density = 1,1540 (2,000–34,000) parasites/μL] and *P. malariae* [parasite density = 2,100 (1,270–3,720) parasites/μL] were identified in this study. There were no mixed infections (Table 2).

The overall point prevalence of asymptomatic malaria by microscopy, RDT and nPCR were 116/500 (23.2%), 156/500 (31.2%), and 184/500 (36.8%), respectively (Fig 2).

The point prevalence of asymptomatic malaria was 98 (19.6%) for males and 86 (17.2%) for females based on nPCR. Using microscopy and RDT, the point prevalence among males was 66 (13.2%) vs 86 (17.2%), respectively, and 50 (10.0%) vs 70 (14.0%) among females, respectively. Upon stratification by age, the point prevalence of asymptomatic malaria was highest in children between 1–2 years of age based on nPCR (16.4%), microscopy (11.2%), and RDT (15.6%), respectively. A similar observation was made after age groups were stratified by gender. Children residing in the community of Worapong (13.2%) presented with the highest point prevalence of asymptomatic malaria, followed by Abira (8.0%), Barekuma and Adankwame (4.4%), and Barekese (4.0%), with the lowest being Esaaso (2.8%). Children in Abira [OR = 4.55, 95% CI (1.90–10.90), p<0.01], Barekese [OR = 4.55, 95% CI (1.58–13.08), p<0.01], and Worapong [OR = 36.0, 95% CI (11.52–112.48), p<0.0001] had significantly higher odds of asymptomatic malaria compared to Barekuma. A similar observation was made

**Table 2. Baseline characteristics of the study population.**

| Variables | Frequency (n = 500) | Percentage (%) |
|---|---|---|
| **Age (years)** | 2.21 ± 1.28* | |
| ≤1 | 62 | 12.4 |
| >1–2 | 236 | 47.2 |
| >2–3 | 54 | 14.8 |
| >3–4 | 76 | 15.2 |
| >4–5 | 52 | 10.4 |
| **Sex** | | |
| Male | 256 | 51.2 |
| Female | 244 | 48.8 |
| **Residence** | | |
| Abira | 88 | 17.6 |
| Adankwame | 94 | 18.8 |
| Barekese | 44 | 8.8 |
| Barekuma | 142 | 28.4 |
| Esaaso | 56 | 11.2 |
| Worapong | 76 | 15.2 |
| **Anaemia** | 368 | 75.6 |
| *Mild* | *276* | *55.2* |
| *Moderate* | *84* | *16.8* |
| *Severe* | *18* | *3.6* |
| **Variable** | **Mean** | *Standard Deviation* |
| **Haemoglobin (g/dL)** | *10.26* | *± 1.46* |
| **Variable** | **Median** | **Interquartile Range** |
| **Parasite density (microscopy)‡** | | |
| *P. falciparum* (parasites/μL) | 11,540 | (2,000–34,000) |
| *P. malariae* (parasites/μL) | 2,100 | (1,270–3,720) |

*Anaemia was defined as haemoglobin level <11 g/dL and graded as mild (10–10.9 g/dL), moderate (7–9.9 g/dL), and severe (<7 g/dL).

when using microscopy and RDT. In addition, children experiencing any anaemia had an increased odds of asymptomatic malaria (**Table 3**).

Nested PCR detected 156 *P. falciparum* cases, of which microscopy identified 88 cases and did not identify 68 cases. RDT detected all 156 cases of *P. falciparum*. All 28 cases of *P. malariae* identified by nPCR were also detected by microscopy. RDT did not detect any *P. malariae* cases (**Table 4**).

RDT presented with a perfect sensitivity (100.0%) specificity (100.0%), accuracy (100.0%), and reliability (100.0%) in detecting *P. falciparum* infection. Microscopy presented with a similar performance with respect to *P. malariae* infection. However, the sensitivity (56.4%), accuracy (85.6%) and reliability (56.4%) of microscopy was attenuated for detecting *P. falciparum* (**Table 5**).

## Discussion

Based on nested PCR, this study reports a high point prevalence (36.8%) of asymptomatic *Plasmodium* infection among a paediatric population in the Atwima Nwabiagya North district of Ghana. Asymptomatic malaria in children under 5 years has been reported in some African countries [33–35]. In Ghana, Crookston, et al. [46] reported an asymptomatic malaria

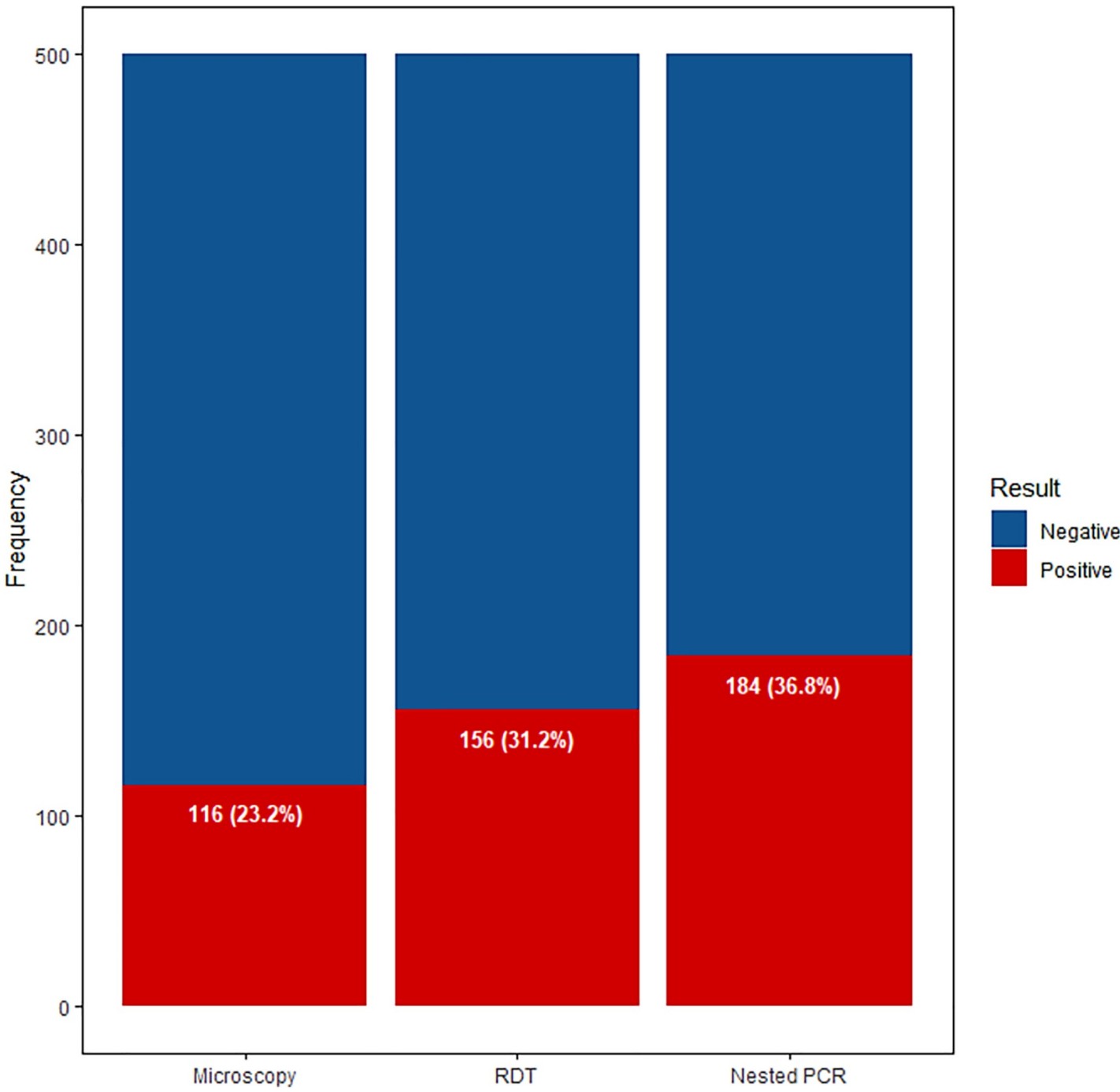

**Fig 2. Prevalence of asymptomatic malaria by microscopy, RDT, and nPCR.**

prevalence of 31.8% based on PCR among children less than five years of age in Kumasi, Ghana which is similar to our study finding. Other studies in Ghana such as those by Dinko, et al. [31] in Kumasi and Danquah, et al. [47] in northern Ghana reported asymptomatic malaria among older children. The prevalence found in the studies by Dinko et al. (76.6%) and Danquah et al. (89.7%) are higher compared to this study and the discrepancy may be linked to the fact that older children tend to be the major asymptomatic carriers of *Plasmodium*

**Table 3. Sociodemographic-stratified point prevalence and odds ratios for asymptomatic malaria infection by microscopy, RDT and nPCR.**

| Variables | Microscopy | | RDT | | nPCR | |
|---|---|---|---|---|---|---|
| | Prevalence | OR (95% CI) | Prevalence | OR (95% CI) | Prevalence | OR (95% CI) |
| **Sex** | | | | | | |
| Female | 50 (10.0) | 1 | 70 (14.0) | 1 | 86 (17.2) | 1 |
| Male | 66 (13.2) | 1.35 (0.75–2.44) | 86 (17.2) | 1.26 (0.74–2.15) | 98 (19.6) | 1.14 (0.68–1.14) |
| **Age group (years)** | | | | | | |
| ≤1 | 12 (2.4) | 1 | 14 (2.8) | 1 | 18 (3.6) | 1 |
| >1–2 | 56 (11.2) | 1.30 (0.48–3.48) | 78 (15.6) | 1.69 (0.67–4.27) | 82 (16.4) | 1.30 (0.55–3.09) |
| >2–3 | 18 (3.6) | 1.34 (0.42–4.30) | 22 (4.4) | 1.45 (0.48–4.35) | 30 (6.0) | 1.67 (0.60–4.46) |
| >3–4 | 22 (4.4) | 1.70 (0.55–5.28) | 26 (5.2) | 1.78 (0.61–5.23) | 32 (6.4) | 1.78 (0.65–4.87) |
| >4–5 | 8 (1.6) | 0.76 (0.19–3.04) | 16 (3.2) | 1.52 (0.47–4.98) | 22 (4.4) | 1.79 (0.60–5.38) |
| *Female* | | | | | | |
| ≤1 | 8 (3.3) | 1 | 12 (4.9) | 1 | 12 (4.9) | 1 |
| >1–2 | 22 (9.0) | 0.84 (0.23–3.04) | 30 (12.3) | 0.71 (0.23–2.24) | 34 (13.9) | 0.85 (0.27–2.64) |
| >2–3 | 10 (4.1) | 1.03 (0.23–4.58) | 14 (5.7) | 0.93 (0.25–3.52) | 20 (8.2) | 1.67 (0.46–6.06) |
| >3–4 | 4 (1.6) | 0.70 (0.11–4.59) | 6 (2.5) | 0.67 (0.13–3.41) | 10 (4.1) | 1.43 (0.32–6.46) |
| >4–5 | 6 (2.5) | 1.05 (0.19–5.76) | 8 (3.3) | 0.89 (0.19–4.11) | 10 (4.1) | 1.25 (0.28–5.53) |
| *Male* | | | | | | |
| ≤1 | 4 (1.6) | 1 | 2 (0.8) | 1 | 3 (2.3) | 1 |
| >1–2 | 34 (13.3) | 2.13 (0.43–10.60) | 48 (18.8) | 7.78 (0.95–63.80) | 48 (18.8) | 2.16 (0.54–8.67) |
| >2–3 | 8 (3.1) | 2.00 (0.30–13.27) | 8 (3.1) | 4.36 (0.42–45.26) | 10 (3.9) | 1.67 (0.31–8.93) |
| >3–4 | 18 (7.0) | 2.91 (0.53–16.09) | 20 (7.8) | 7.50 (0.84–66.86) | 22 (8.6) | 2.44 (0.54–11.03) |
| >4–5 | 2 (0.8) | 0.46 (0.04–5.79) | 8 (3.1) | 5.33 (0.51–56.24) | 12 (4.7) | 2.86 (0.53–15.47) |
| **Anaemic status** | | | | | | |
| Non-anaemic | 8 (1.6) | 1 | 16 (3.2) | 1 | 18 (3.6) | 1 |
| Anaemic | 108 (21.6) | 5.70 (1.97–16.48)** | 140 (28.0) | 3.90 (1.75–8.67)** | 166 (33.2) | 4.54 (2.11–9.71)*** |
| **Residence** | | | | | | |
| Barekuma | 16 (3.2) | 1 | 20 (4.0) | 1 | 22 (4.4) | 1 |
| Barekese | 8 (1.6) | 1.75 (0.47–6.48) | 10 (2.0) | 1.79 (0.54–5.96) | 20 (4.0) | 4.55 (1.58–13.08)** |
| Adankwame | 16 (3.2) | 1.62 (0.56–4.65) | 18 (3.6) | 1.45 (0.54–3.88) | 22 (4.4) | 1.67 (0.66–4.23) |
| Esaaso | 0 (0.0) | - | 10 (2.0) | 1.33 (0.41–4.30) | 14 (2.8) | 1.82 (0.62–5.30) |
| Abira | 28 (5.6) | 3.68 (1.39–9.71)** | 40 (8.0) | 5.08 (2.08–12.43)*** | 40 (8.0) | 4.55 (1.90–10.90)** |
| Worapong | 48 (9.6) | 13.50 (5.03–36.25)*** | 58 (11.9) | 19.66 (7.21–53.60)*** | 66 (13.2) | 36.00 (11.52–112.48)*** |

**; Significant at p<0.01

***; Significant at p<0.0001

compared to younger children [47, 48], possibly as a result of protective immunity acquired over the years.

We also found asymptomatic malaria to be more prevalent among male children compared to females, similar to the findings of Golassa et al. in Ethiopia [29]. The higher point prevalence

**Table 4. Comparison of microscopy and RDT with nPCR for the detection of asymptomatic malaria infection.**

| Methods | *P. falciparum* | *P. malariae* | Negative |
|---|---|---|---|
| Microscopy | 88 | 28 | 384 |
| RDT | 156 | n/a | 344 |
| nPCR | 156 | 28 | 316 |

**Table 5. Diagnostic performance of microscopy and RDT in detecting asymptomatic malaria infection.**

| Methods | Microscopy | | RDT |
|---|---|---|---|
| | *P. falciparum* | *P. malariae* | *P. falciparum* |
| Sensitivity (95% CI) | 56.4 (48.3–64.3) | 100.0 (87.7–100.0) | 100.0 (97.7–100.0) |
| Specificity (95% CI) | 100.0 (98.8–100.0) | 100.0 (98.8–100.0) | 100.0 (98.9–100.0) |
| PPV | 100.0 | 100.0 | 100.0 |
| NPV | 82.3 | 100.0 | 100.0 |
| TP | 88 | 28 | 156 |
| TN | 316 | 316 | 344 |
| FP | 0 | 0 | 0 |
| FN | 68 | 0 | 0 |
| Accuracy (%) | 85.6 | 100.0 | 100.0 |
| AUC (%) | 78.2 (74.2–81.8) | 100.0 (98.9–100.0) | 100.0 (99.3–100.0) |
| Reliability (%) | 56.4 | 100.0 | 100.0 |

nPCR was used as the reference

of asymptomatic malaria among male children compared to females could be explained by the fact that, in Ghana, males are more exposed to both daytime and nighttime outdoor activities than females and thus, at higher risk of mosquito bites compared to females who are usually indoors. Our findings also indicate a strong association between presence of asymptomatic malaria and anaemia as consistent with studies by Crookston et al. [46], Verhoef et al. [49], and Stoltzfus et al. [50]. This study also presents information on the point prevalence of asymptomatic malaria in six different communities, providing data on potential hotspots for malaria screening and treatment. Children residing in Worapong presented with the highest point prevalence of asymptomatic malaria, followed by children in Abira. The higher point prevalence in Worapong compared to the other communities could be due to the relative accessibility of the communities or the agricultural practices of the specific communities. Barekese and Adankwame are economically diverse communities and on a main through-road, whereas Worapong is very difficult to get access due to poor roads and flooded rice field farming provides habitat for malaria transmitting mosquitoes. Nonetheless, owing to the fact that malaria-related morbidity and mortality is high among children and the propensity of asymptomatic malaria transitioning into clinical malaria, the children in these communities should be given substantial precedence during national and regional malaria surveillance exercises.

Due to the high point prevalence of asymptomatic malaria in Ghana [31, 32, 48], and the possibility that people with asymptomatic malaria infection may serve as a significant reservoir, transmitting *Plasmodium* to uninfected *Anopheles* mosquitoes [28, 29], prompt and accurate diagnosis of asymptomatic *Plasmodium* infection is crucial. In this study, despite the high specificity, microscopy presented with poor sensitivity and reliability for the detection of asymptomatic *P. falciparum* compared to nPCR as consistent with previous reports [14, 19, 27, 29, 44]. Taken together, these findings indicate that a negative result by microscopy does not exclude the presence of malaria infection since a substantial number of false negatives were associated with microscopy for *P. falciparum*. Microscopy thus seems to have an inherent limitation for asymptomatic *P. falciparum* detection, with proficiency of technicians and microscopists likely to be a significant contributor. This underscores the need for continued refinement through constant re-training to sharpen microscopists' ability to detect malaria cases especially at low parasite densities, since refresher training has been reported to significantly improve the

diagnostic accuracy of parasitological diagnosis of malaria by microscopy [51]. Meanwhile, the high specificity and positive predictive value of microscopy in detecting malaria parasites, regardless of the species, suggest that positive malaria microscopy is a good confirmation of malaria, regardless of the presence of symptoms. Thus, a positive result from microscopy could be trusted as the presence of the *Plasmodium* infection and anti-parasitic therapy should be guided by the species identified.

Strikingly, despite the possibility of false negatives due to *pf*HRP-2 gene deletions [21–23] and the persistence of *pf*HRP-2 antigens in circulation even after parasite clearance, which may increase the incidence of false positive results [24], malaria diagnosis by RDT provided a better estimate of asymptomatic *P. falciparum* infection, with a perfect sensitivity, specificity, accuracy and reliability compared to microscopy when nPCR is used as the reference. Indeed, RDT was able to correctly classify all *P. falciparum* cases detected by nPCR. This may be due to the fact that the RDT used in this study detects *pf*HRP-2 antigen and not malaria parasites, affording it an added advantage over microscopy through the detection of antigens produced in very low parasite densities below the detection threshold of microscopy. This finding strongly suggests that detection of *pf*HRP-2 by RDT accurately identifies *P. falciparum* infection in asymptomatic children and is an indication for anti-malarial therapy. However, it is worthy of note that there is the possibility that the *pf*HRP-2 antigen may be persistent in the blood, even in the absence of viable parasites. Our study excluded children who had received anti-malaria therapy in the prior 14 days. It is possible that if the study had included children who have had such therapy within the prior 14 days, we would have identified children who had persistent *pf*HRP-2 antigenemia but who were no longer parasitemic. Thus, interpretation should be done with caution. Additionally, it should be noted that the widely used RDT for malaria diagnosis in Ghana is *pf*HRP-2-based RDT which detects on *P. falciparum* but not *P. malariae*. The choice of this RDT is attributed to the relatively higher point prevalence of *P. falciparum* and its associated clinical significance in Ghana compared to the *P. malariae* which causes less severe clinical outcomes [52]. In other words, a negative *pf*HRP-2-based RDT does not exclude infection with non-falciparum malaria species.

Although the finding of this study and several other reports point to the fact that RDT should be used as a surrogate to microscopy due to the low sensitivity of microscopy, it is worthy of note that, microscopy allows for the quantification and calculation of malaria parasite densities, a function which RDT cannot be used to assess. Thus, microscopy should not be abandoned. We, however, recommend consistent re-training of malaria microscopists in the region to enhance their *Plasmodium* detection skills and abilities. Moreover, despite the high sensitivity and specificity, PCR is still expensive which limits its usefulness in routine malaria diagnosis.

### Study strengths and limitations

The strength of the study is in the reporting of the point prevalence of asymptomatic malaria infection among children less than 5 years old in the northern sector of Ghana. We also highlight potential hotspots for malaria screening and treatment during national and regional malaria surveillance exercises. The study also corroborates previous reports on the usefulness of molecular detection methods for asymptomatic malaria diagnosis. We showed that, in the absence of PCR, RDT performs better in the diagnosis of asymptomatic malaria caused by P. falciparum among children compared to microscopy.

This study is however limited the fact that we used only a single brand of malaria RDT; the prevalence may not be the same when other commercially available test kits are used. Also, data on the use of long-lasting insecticidal nets was unavailable. Additionally, the study was

conducted in a peri-urban setting and the findings may not be generalizable to other areas. Thus, we recommend that further studies be conducted in the larger population.

## Conclusions

The findings of this study indicate a high point prevalence of *Plasmodium* infection and anemia among asymptomatic children in Atwima Nwabiagya North district of Ghana. Since Ghana remains in the control stage, there is the exigent need for effort intensification through detection of asymptomatic malaria, using highly sensitive diagnostic tools, in order to reach the pre-eradication stage of malaria. The use of microscopy for *Plasmodium* detection in children who are asymptomatic presents several challenges. However, in the absence of the more sensitive PCR, the use RDT provides substantial diagnostic sensitivity and reliability.

## Supporting information

**S1 File. Author information TDickerson malaria RDT Ghana.**
(DOCX)

## Acknowledgments

The authors are grateful to all children and their parents/guardians who participated in the study. Authors are also grateful to the Student Learning Abroad Group of 2015 for their support as research assistants. A part of the work has been presented as a poster at the 2016 American Society for Microbiology General Meeting.

## Author Contributions

**Conceptualization:** Alex Owusu-Ofori, Daniel Ansong, Rebecca Buxton, Scott Benson, Alex Osei-Akoto, Joseph Marfo Boaheng, Ty Dickerson.

**Data curation:** Bismark Okyere, Collins Adjei, Evans Xorse Amuzu.

**Formal analysis:** Eddie-Williams Owiredu.

**Investigation:** Bismark Okyere, Scott Benson, Collins Adjei, Evans Xorse Amuzu, Joseph Marfo Boaheng.

**Methodology:** Alex Owusu-Ofori, Daniel Ansong, Rebecca Buxton, Scott Benson, Alex Osei-Akoto, Evans Xorse Amuzu, Joseph Marfo Boaheng, Ty Dickerson.

**Project administration:** Evans Xorse Amuzu, Joseph Marfo Boaheng.

**Supervision:** Alex Owusu-Ofori, Daniel Ansong, Rebecca Buxton, Scott Benson, Ty Dickerson.

**Validation:** Bismark Okyere.

**Visualization:** Eddie-Williams Owiredu.

**Writing – original draft:** Bismark Okyere, Eddie-Williams Owiredu.

**Writing – review & editing:** Bismark Okyere, Alex Owusu-Ofori, Daniel Ansong, Rebecca Buxton, Scott Benson, Alex Osei-Akoto, Eddie-Williams Owiredu, Evans Xorse Amuzu, Joseph Marfo Boaheng, Ty Dickerson.

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
