## [Decision Letter · Decision Letter 0]

18 Feb 2020

PONE-D-20-02713

Asymptomatic Plasmodium infection in a paediatric population in Ghana: prevalence and comparison of microscopy, rapid diagnostic tests and nested PCR

PLOS ONE

Dear  Dr. DICKERSON,

Thank you for submitting your manuscript for review to PLoS ONE. After careful consideration, we feel that your manuscript will likely be suitable for publication if the authors revise it to address specific point raised by the reviewers. According to reviewers, there are still some areas where further improvements would be of substantial benefit to the readers, including study design and results.  A major concern raised by the reviewer #1 was about the PCR protocol used as reference, as it cannot identify *P.vivax* infections (potential bias). Consequently, the tittle of the manuscript should be properly adjusted as suggested by reviewer #2.  Finally, the manuscript should benefit from a proofreading.

We would appreciate receiving your revised manuscript by March 30. To enhance the reproducibility of your results, we recommend that if applicable you deposit your laboratory protocols in protocols.io, where a protocol can be assigned its own identifier (DOI) such that it can be cited independently in the future. For instructions see: http://journals.plos.org/plosone/s/submission-guidelines#loc-laboratory-protocols

We look forward to receiving your revised manuscript.

Kind regards,

Luzia Helena Carvalho, Ph.D.

Academic Editor

PLOS ONE

Journal Requirements:

1. Thank you for including your funding statement;"The funders had no role in study design, data collection and analysis, decision to publish, or preparation of the manuscript."

Please provide an amended Funding Statement that declares *all* the funding or sources of support received during this specific study (whether external or internal to your organization) as detailed online in our guide for authors at http://journals.plos.org/plosone/s/submit-now.  

Please state what role the funders took in the study.  If any authors received a salary from any of your funders, please state which authors and which funder. If the funders had no role, please state: "The funders had no role in study design, data collection and analysis, decision to publish, or preparation of the manuscript."

2. 

PLOS ONE now requires that authors provide the original uncropped and unadjusted images underlying all blot or gel results reported in a submission’s figures or Supporting Information files. This policy and the journal’s other requirements for blot/gel reporting and figure preparation are described in detail at https://journals.plos.org/plosone/s/figures#loc-blot-and-gel-reporting-requirements and https://journals.plos.org/plosone/s/figures#loc-preparing-figures-from-image-files. When you submit your revised manuscript, please ensure that your figures adhere fully to these guidelines and provide the original underlying images for all blot or gel data reported in your submission. See the following link for instructions on providing the original image data: https://journals.plos.org/plosone/s/figures#loc-original-images-for-blots-and-gels.

3. Please amend the manuscript submission data (via Edit Submission) to include author Eddie-Williams Owiredu

Reviewers' comments:

Reviewer's Responses to Questions

**Comments to the Author**

1. Is the manuscript technically sound, and do the data support the conclusions?

Reviewer #1: Yes

Reviewer #2: Yes

2. Has the statistical analysis been performed appropriately and rigorously? 

Reviewer #1: Yes

Reviewer #2: Yes

3. Have the authors made all data underlying the findings in their manuscript fully available?

Reviewer #1: Yes

Reviewer #2: Yes

4. Is the manuscript presented in an intelligible fashion and written in standard English?

Reviewer #1: Yes

Reviewer #2: Yes

5. Review Comments to the Author

Reviewer #1: The submission by Dickerson and colleagues describes a comparison between light microscopy, rapid diagnostic tests and nPCR (as reference) to assess malaria asymptomatic infection in children less than 5 years. The methodology globally sounds good and is clearly reproductible. Their results are in line with relative literature.

The paper is clearly written and the discussion is thoughtful as they point out limits of their work.

This paper should, as the authors emphasize, galvanize far more intensive surveillance for these asymptomatic bearers.

My minor comments are detailed below.

Line 65: there is no reference between brackets

Lines 120 -121: Have you collected any information about LLIN use?

Lines 160-161: the sentence ‘Only P. 161 falciparum and P. malariae were identified in this study. There were no mixed infections by nPCR’ should be moved to Results section.

One question: The PCR you used cannot diagnose P. vivax. As you consider this PCR as the reference, I think that there is a little bias. Could you explain this choice ?

Reviewer #2: General comments;

This manuscript addresses an important area which is highly relevant to malaria control in areas which are transitioning from holo/hyper-endemic to hypo-endemic transmission of malaria. However, the manuscript needs minor revision to better present the key questions addressed, the findings, and clearly show the novelty.

1. Title: Well written but it can be rephrased to give a clear message. The authors were assessing the prevalence of asymptomatic P. falciparum infection in paediatric population in Ghana, but also compared the diagnostic tools, so they should rephrase these two sentences in order to capture reader's attention just from the title.

2. Background: The concept of malaria prevalence in endemic areas and in the country where the study was conducted needs to be thoroughly explained. The authors should clearly indicate the relevance of conducting this study. A number of this kind of studies have been conducted in Ghana, I suggest that, the authors should clearly show that the study was conducted in (Atwima Nwabiagya) Northern district in Ghana, not just Ghana in general.

3. Results: The authors need to check their results the way they have reported them. What I know and from several reports, asymptomatic malaria occurs in children above 5 years of age and not less than 5 because of immunity. But their results indicate children of 1-2 years appear to be asymptomatic compared to other age groups. They need to justify this.

- Figures are also missing in the document that I review.

- Parasite density is also important to be shown

- There should be parasite density of the results obtained by microscopy. This will add value and will be a relevant information as well.

4. Discussion: this section is poorly written. There are some facts that need to be considered in order to reflect the rationale of the study and the findings. Authors should handle all issues raised and re-write this section accordingly.

5. Study strength and limitations: This section is missing. They should have clearly pointed this out.

6. Other minor comments

The authors should check the language throughout the manuscript and correct typos. I have done my comments in the manuscript in track changes.

6. PLOS authors have the option to publish the peer review history of their article (what does this mean?). If published, this will include your full peer review and any attached files.

Reviewer #1: No

Reviewer #2: Yes: Dr. Deborah Sumari

---

## [Author Response · Author response to Decision Letter 0]

14 Apr 2020

The Editor,

PLOS ONE

Dear Sir/ Madam,

RESPONSE TO REVIEWERS

The authors appreciate the timely and scrupulous review of our manuscript (PONE-D-20-02713). The authors note seventeen (17) comments in total. Kindly find below the responses to the reviewers’ comments. Tracked changes have been employed to highlight manuscript texts revised per the reviewers’ recommendations. Revised text indicated in this response are in “quotation marks”.

Reviewer #1: 

Line 65: there is no reference between brackets

Response 4: Reference #4 has been added to the appropriate text

Lines 120 -121: Have you collected any information about LLIN use?

Response: Thank you for your comment. Because objective of the study is to determine the prevalence of asymptomatic malaria infection and evaluate the performance of RDT, microscopy and PCR for malaria diagnosis, we did not collect other data such as use of LLIN. However, we have included it as a limitation of the study (lines 279-282).

“This study is however limited the fact that we used only a single brand of malaria RDT; the prevalence may not be the same when other commercially available test kits are used. Also, data on the use of long-lasting insecticidal nets was unavailable. Additionally, the study was conducted in a peri-urban setting and the findings may not be generalizable to other areas. Thus, we recommend that further studies be conducted in the larger population.”

Lines 160-161: the sentence ‘Only P. 161 falciparum and P. malariae were identified in this study. There were no mixed infections by nPCR’ should be moved to Results section.

Response: Thank you for your comment. The statement has been moved to the results section (lines 178-179)

“Only P. falciparum [parasite density= 1,1540 (2,000- 34,000) parasites/µL] and P. malariae [parasite density= 2,100 (1,270- 3,720) parasites/µL] were identified in this study. There were no mixed infections”

One question: The PCR you used cannot diagnose P. vivax. As you consider this PCR as the reference, I think that there is a little bias. Could you explain this choice?

Response: Thank you for your comment. The PCR primers were selected based on the occurrence/prevalence of the different species of Plasmodium in Ghana. It is known that there are currently five major clinically relevant Plasmodium species, namely, P. falciparum, P. ovale, P. malariae, P. vivax and P. knowlesi. However, only P. falciparum, P. ovale, P. malariae have been documented to occur in Ghana (https://www.who.int/malaria/publications/country-profiles/profile_gha_en.pdf?ua=1/ and https://www.severemalaria.org/countries/ghana and http://www.ghanahealthservice.org/downloads/GHS_Antimalaria_drug_policy.pdf ). We did not include P. vivax because reports of its infection in the Ghana is rare.

Reviewer #2:

1. Title: Well written but it can be rephrased to give a clear message. The authors were assessing the prevalence of asymptomatic P. falciparum infection in paediatric population in Ghana, but also compared the diagnostic tools, so they should rephrase these two sentences in order to capture reader's attention just from the title.

Response: Thank you for your comment. We have rephrased the title to clearly highlight the prevalence and comparison of test methods (lines 1-3).

“Prevalence of asymptomatic Plasmodium infection and the comparison of microscopy, rapid diagnostic test and nested PCR for the diagnosis of asymptomatic malaria among children under 5 years in Ghana”

2. Background: The concept of malaria prevalence in endemic areas and in the country where the study was conducted needs to be thoroughly explained. The authors should clearly indicate the relevance of conducting this study. A number of this kind of studies have been conducted in Ghana, I suggest that, the authors should clearly show that the study was conducted in (Atwima Nwabiagya) Northern district in Ghana, not just Ghana in general.

Response: Thank you for your comment. The relevance of the study has been clarified and (Atwima Nwabiagya) Northern district as the study site has been highlighted (lines 96-101)

“Although a number of studies on asymptomatic malaria in older children have been conducted across Ghana [30-32] and children under 5 in neighboring African countries [33-35], there remains a dearth of published data on asymptomatic malaria in children under 5 years in Ghana, particular in the northern sectors of the country where adequate health facilities are wanting. This study assessed the prevalence of asymptomatic malaria infection and evaluated the performance of malaria RDT, light microscopy and nested PCR (nPCR) for the diagnosis of asymptomatic malaria infection in children under 5 years old in Atwima Nwabiagya North district, Ghana.

3. Results: The authors need to check their results the way they have reported them. What I know and from several reports, asymptomatic malaria occurs in children above 5 years of age and not less than 5 because of immunity. But their results indicate children of 1-2 years appear to be asymptomatic compared to other age groups. They need to justify this.

- Figures are also missing in the document that I review.

- Parasite density is also important to be shown

- There should be parasite density of the results obtained by microscopy. This will add value and will be a relevant information as well.

Response: Thank you very much for your comment. The data has been rechecked and the results remained the same. In this study, simple random sampling procedure was used to recruit asymptomatic children. The higher prevalence of asymptomatic malaria among 1-2 years old children could be due to the greater number/size of 1-2 year aged children obtained in this study. This could also highlight the distribution of asymptomatic malaria cases in children under 5 in the region. The figures were included during the submission. Parasite densities have been included (table 1 and lines 178-179).

“Only P. falciparum [parasite density= 1,1540 (2,000- 34,000) parasites/µL] and P. malariae [parasite density= 2,100 (1,270- 3,720) parasites/µL] were identified in this study. There were no mixed infections”

4. Discussion: this section is poorly written. There are some facts that need to be considered in order to reflect the rationale of the study and the findings. Authors should handle all issues raised and re-write this section accordingly.

Response: Thank you for your comment. The discussion section has been revised, taking into consideration all comments attached (line 216-227)

“Based on PCR, this study reports a high prevalence (36.8%) of asymptomatic Plasmodium infection among a paediatric population in the Atwima Nwabiagya North district of Ghana. Asymptomatic malaria in children under 5 years has been reported in some African countries [33-35]. In Ghana, Crookston, et al. [46] reported an asymptomatic malaria prevalence of 31.8% based on PCR among children less than five years of age in Kumasi, Ghana which is similar to our study finding. Other studies in Ghana such as those by Dinko et al. [31] in Kumasi and Danquah et al. [47] in northern Ghana reported asymptomatic malaria among older children. The prevalence found in the studies by Dinko et al. (76.6%) and Danquah et al. (89.7%) are higher compared to this study and the discrepancy may be linked to the fact that older children tend to be the major asymptomatic carriers of Plasmodium compared to younger children [47, 48], possibly as a result of protective immunity acquired over the years.

We also found asymptomatic malaria to be more prevalent among male children compared to females, similar to the findings of Golassa et al. in Ethiopia [29]. The higher prevalence of asymptomatic malaria among male children compared to females could be explained by the fact that, in Ghana, males are more exposed to both daytime and nighttime outdoor activities than females and thus, at higher risk of mosquito bites compared to females who are usually indoors …”

5. Study strength and limitations: This section is missing. They should have clearly pointed this out.

Response: Thank you for your comment. We have included a section for study strength and limitations (line 273-282)

“Study strengths and limitations

The strength of the study is in the reporting of the prevalence of asymptomatic malaria infection among children less than 5 years old in the northern sector of Ghana. We also highlight potential hotspots for malaria screening and treatment during national and regional malaria surveillance exercises. The study also corroborates previous reports on the usefulness of molecular detection methods for asymptomatic malaria diagnosis. We showed that, in the absence of PCR, RDT performs better in the diagnosis of asymptomatic malaria among children compared to microscopy.

This study is however limited the fact that we used only a single brand of malaria RDT; the prevalence may not be the same when other commercially available test kits are used. Also, data on the use of long-lasting insecticidal nets was unavailable. Additionally, the study was conducted in a peri-urban setting and the findings may not be generalizable to other areas. Thus, we recommend that further studies be conducted in the larger population.”

6. Other minor comments

The authors should check the language throughout the manuscript and correct typos. I have done my comments in the manuscript in track changes.

Response: Thank you very much for your comment. The language and typos have been reviewed and revised and all comments included as tracked changes have been addressed.

Miscellaneous Author Responses: 

• The 2010 Population and Housing Census of Ghana is the latest published data. A newer census is scheduled to take place in 2020.

• The authors initially chose to use the term point prevalence for malaria as our study reports the prevalence of malaria cases among individual asymptomatic children at a single point in time. Whereas point prevalence addresses the question of whether a person currently has malaria, period prevalence addresses the question of whether that subject had malaria at any time during the period under investigation. Since the study did not test individual children for malaria at multiple times over a defined period of time, we are unable to report the period prevalence for asymptomatic malaria infection. 

Thank you once again for the timely review and scrupulous of our manuscript. Looking forward to hear favorably from you.

Sincerely,

Ty Dickerson

Corresponding author

---

## [Decision Letter · Decision Letter 1]

24 Apr 2020

Point prevalence of asymptomatic Plasmodium infection and the comparison of microscopy, rapid diagnostic test and nested PCR for the diagnosis of asymptomatic malaria among children under 5 years in Ghana

PONE-D-20-02713R1

Dear Dr.  DICKERSON,

We are pleased to inform you that your manuscript has been judged scientifically suitable for publication and will be formally accepted for publication once it complies with all outstanding technical requirements.

With kind regards,

Luzia Helena Carvalho, Ph.D.

Academic Editor

PLOS ONE

Additional Editor Comments (optional):

Reviewers' comments:

Reviewer's Responses to Questions

**Comments to the Author**

1. If the authors have adequately addressed your comments raised in a previous round of review and you feel that this manuscript is now acceptable for publication, you may indicate that here to bypass the “Comments to the Author” section, enter your conflict of interest statement in the “Confidential to Editor” section, and submit your "Accept" recommendation.

Reviewer #1: All comments have been addressed

2. Is the manuscript technically sound, and do the data support the conclusions?

Reviewer #1: Yes

3. Has the statistical analysis been performed appropriately and rigorously? 

Reviewer #1: Yes

4. Have the authors made all data underlying the findings in their manuscript fully available?

Reviewer #1: Yes

5. Is the manuscript presented in an intelligible fashion and written in standard English?

Reviewer #1: Yes

6. Review Comments to the Author

Reviewer #1: We are thankful to the authors and approve this updated version of the manuscript. The authors replied to all of our comments.

7. PLOS authors have the option to publish the peer review history of their article (what does this mean?). If published, this will include your full peer review and any attached files.

Reviewer #1: Yes: Dieudonné M. Mvumbi

---

## [Editor Report · Acceptance letter]

8 Jul 2020

PONE-D-20-02713R1 

Point prevalence of asymptomatic Plasmodium infection and the comparison of microscopy, rapid diagnostic test and nested PCR for the diagnosis of asymptomatic malaria among children under 5 years in Ghana 

Dear Dr. DICKERSON:

I'm pleased to inform you that your manuscript has been deemed suitable for publication in PLOS ONE. Congratulations! Your manuscript is now with our production department. 

Kind regards, 

on behalf of

Dr. Luzia Helena Carvalho 

Academic Editor

PLOS ONE